# Stability of Biological Activity of Frozen β-lactams over Time as Assessed by Time-Lapsed Broth Microdilutions

**DOI:** 10.3390/antibiotics8040165

**Published:** 2019-09-25

**Authors:** Gleice Leite, Alexander Lawandi, Matthew P. Cheng, Todd Lee

**Affiliations:** 1Research Institute of the McGill University Health Centre, Montréal, QC H4A 3J1, Canada; gleice.leite@muhc.mcgill.ca (G.L.); matthew.cheng@mail.mcgill.ca (M.P.C.); todd.lee@mcgill.ca (T.L.); 2Division of Infectious Diseases, Department of Medicine, McGill University Health Centre, Montréal, QC H4A 3J1, Canada; 3McGill Interdisciplinary Initiative in Infection and Immunity, McGill University, Montréal, QC H3A 1Y2, Canada

**Keywords:** Beta-lactams, antimicrobial stability, frozen drugs, antimicrobial

## Abstract

To evaluate the antimicrobial agent’s stability stored at −80 °C, six β-lactams (meropenem, ertapenem, imipenem, ceftriaxone, ceftazidime, and piperacillin-tazobactam) were studied using the broth microdilution (BMD). The minimum inhibitory concentration (MIC) remained accurate with the same amount of frozen drug for at least six months. Thereafter, there was a diminishing drug concentration due to instability. At this temperature, most β-lactams can be frozen as a stock concentration for up to six months without a significant loss in antibiotic activity.

Antibiotic susceptibility testing (AST) is routinely performed in both clinical and research laboratories yielding important data for clinical and epidemiological purposes [1]. For most organisms, the gold standard in AST remains the broth microdilution assay (BMD), a method that determines the minimum inhibitory concentrations (MICs) of each tested antibiotic against the bacterial isolate of interest [2]. While usually reproducible, there can be important variability in any MIC measurement. This may occur through biological variation such as “MIC creeps” as well as assay variation in terms of inoculum preparation, media, incubation temperature and time, and variation between laboratories [3]. In order for the MIC to be determined, the concentrations and the biological activity of the antibiotics used in the assays must also be predictably accurate such that variations in the MIC are not reflections of antibiotic degradation or loss of activity through other mechanisms. To the best of our knowledge, the stability of biological activity of reconstituted β-lactams antibiotics stored at –80 °C has not been determined. Therefore, the purpose of this study was to evaluate the stability of these antimicrobial agents in terms of their biological activity for use in BMD assays that were carried out periodically over a ten-month period.

Six β-lactams were purchased from different pharmaceutical companies: meropenem trihydrate 1 g (Fresenius Kabi, Toronto, ON, CA), ertapenem sodium 1 g (Merck Canada Inc., Kirkland, QC, CA), imipenem/cilastatin sodium 500 mg/500 mg (Sandoz Canada Inc., Boucherville, QC, CA), ceftriaxone sodium 2 g, ceftazidime pentahydrate 2 g (Sterimax Inc., Oakville, ON, CA), and piperacillin sodium salt 1 g (Sigma-Aldrich, St Louis, MO, USA) combined with Tazobactam sodium salt 250 mg (Cayman Chemical, Ann Arbor, MI, USA). For the meropenem, ertapenem, imipenem, ceftriaxone and ceftazidime, these were clinical grade reagents. For the piperacillin and tazobactam, the purity was >95% [4,5,6,7,8,9,10] and they were dissolved according to the manufacturer’s instructions and stock solutions were prepared in sterile water and stored in aliquots of 2 mL at −80 °C (Table 1). The temperature regulation of the freezer was routinely monitored through daily temperature checks and the temperature was maintained at −80 ± 2 °C throughout the study period without any recorded deviation.

Broth microdilutions were repeatedly performed on a group of 24 KPC producing *Klebsiella pneumoniae* over a period of six months using the standard technique [3] and aliquots of the antibiotics were thawed at room termpature and diluted to the working concentration ranges using sterile water. As a control for each experiment, *Escherichia coli* ATCC 25922 was used as the recommended quality control strain to use when performing AST for *Enterobacteriaceae* according to the CLSI M100 [3].

The MICs remained accurate with the same amount of frozen drug for at least six months. However, thereafter, there was a loss of biological activity as evidenced by what appeared to be rising MICs (see Table 2 for the results for the reference strain with well characterized MICs) but was, in fact diminishing drug biological activity due to instability under the storage conditions. This was demonstrated by confirming that the MICs remained within the expected reference range when the AST of the quality control strains were verified by disc diffusion assay and by Vitek2 AST testing. This was then further confirmed by repeating the BMDs using frenshly prepared new drug lots. 

In conclusion, it appears that most β-lactams can be frozen as a stock concentration for up to six months at −80 °C without a significant loss in antibiotic activity. This can allow for cost savings via bulk purchasing and alioquoting in the laboratory and could also have clinical applications with unused doses in the clinical pharmacy. Beyond six months caution is advised and the biological activity of the antibiotics need verification.

## Figures and Tables

**Table 1 antibiotics-08-00165-t001:** Stock concentrations of the beta-lactams.

Antibiotics	Stock Concentrations (g/L)
Meropenem	11.2
Ertapenem	16.5
Imipenem	5.7
Ceftriaxone	25.7
Ceftazidime	12.9
Piperacillin	16
Tazobactam	0.25

**Table 2 antibiotics-08-00165-t002:** Antibiotic activity for 6 β-lactams stored at −80 °C.

Duration of Antibiotic Storage at −80 °C (days)	MIC of β-Lactams Tested (mg/L)					
	**MER**	**ETP**	**IMI**	**CTX**	**CTZ**	**^b^ PTZ**
0	0.016	0.008	0.12	0.06	0.25	1/4
14	0.016	0.008	0.12	0.06	0.25	4/4
49	0.016	0.004	0.12	0.06	0.25	4/4
179	0.016	0.008	0.25	0.12	0.5	2/4
286	0.25	0.06	2	0.5	2	8/4
^a^ MIC QC ranges (mg/L)						
*E. coli* ATCC25922	0.008–0.016	0.004–0.016	0.06–0.25	0.03–0.12	0.06–0.5	1/4–4/4

^a^ MIC Quality Control ranges according to CLSI, 2018 [3]. These are the expected MIC ranges for the different antibiotics for the *E. coli* ATCC25922 strain and are the same for each time point of antibiotic storage. Abbreviations: MER-meropenem, ETP-ertapenem, IMI-imipenem, CTX-ceftriaxone, CTZ-ceftazidime, PTZ-piperacillin-tazobactam.CLSI M100 [3] suggests using a fixed concentration of tazobactam at 4 mg/L for ^b^ PTZ broth microdilution assays.

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
