# Peer review of "Stability of Biological Activity of Frozen β-lactams over Time as Assessed by Time-Lapsed Broth Microdilutions"

_antibiotics, 2019, doi:10.3390/antibiotics8040165_

Round 1

Reviewer 1 Report

Not applicable

Author Response

Dear reviewer,

Thank you for the opportunity to revise our manuscript.

Sincerely,

Gleice Leite

Reviewer 2 Report

In this presentation, the authors study the stability of six β-lactams in water solution stored in minus 80 oC. They use broth micro dilution assay to measure the minimum inhibitory concentrations based on which they conclude about antibiotic stability as a time course. The work is of low interest and there are many points which need further declaration:

which is the purity of antibiotics and the chemical forms (salts, or other additives). Additionally, which is the rate of compounds in the case of PTZ? table 2 is not clear and conclusive: i) Units are missing, ii) statistical analysis is also missing iii) control values are also missing for all time points and iv) ratio values in the case of PTZ explanation are necessary to satisfy all readers of the journal. 

Author Response

Dear reviewer,

Thank you for the opportunity to revise our manuscript. We also greatly appreciate your complimentary comments and suggestions. We have highlighted the changes within the manuscript.

Please find below a point-by-point response. We hope that you find our responses satisfactory and acceptable for publication.

Comments and Suggestions for Authors

In this presentation, the authors study the stability of six β-lactams in water solution stored in minus 80 oC. They use broth microdilution assay to measure the minimum inhibitory concentrations based on which they conclude about antibiotic stability as a time course. The work is of low interest and there are many points which need further declaration:

We thank you for your insightful commentary and points.

Which is the purity of antibiotics and the chemical forms (salts, or other additives)?

We agree with the reviewer and we have added products information as reference material, making available their product monograph links in the reference topic.

Which is the rate of compounds in the case of PTZ?

It is not clear for us what the reviewer means about the rate of a compound. If the ratio is meant, we followed CLSI recommendations and kept the fixed concentration of tazobactam while titrating the concentration of piperacillin.

Table 2 is not clear and conclusive:

a. Units are missing.

We thank the reviewer for pointing out this oversight. We have now added the units to Table 2.

b. Statistical analysis is also missing.

As the control values for the MIC for the reference strain include an acceptable range that accounts for acceptable measurement error, we do not think any further analysis is required. We had consistently rising MICs across all strains tested over time. The consistency of the trend would not suggest an error in the measurement. Furthermore, as our results returned to the expected range when using fresh antibiotics, we feel strongly that the results require no statistical analysis.

c. Control values are also missing for all time points.

The control values requested are reported at the bottom of Table 2. (MIC Quality Control ranges according to CLSI, 2018).

d. Ratio values in the case of PTZ explanation are necessary to satisfy all readers of the journal.

As suggested by the reviewer, we have added an explanation to the table 2 footnotes according to CLSI recommendations. Briefly, when performing broth microdilutions utilizing beta-lactam/beta-lactamase inhibitor combinations, the concentration of the inhibitor is fixed while the beta-lactam is titrated. Consequently, for Piperacillin-Tazobactam, the tazobactam concentration was always 4 mg/L while the piperacillin concentration was titrated.

Round 2

Reviewer 2 Report

The manuscript has been improved. Two minor points:

TABLE II need two more corrections,

No of observations may be replaced with time of observation or day of Observation and  a footnote in MICQC may be added to specify the time for which MIC values are referred to (dayo to day286 or any different). 

Author Response

We thank the reviewer for their revisions. We have amended the Table as suggested. We have removed "No of Observations" and replaced it with "Duration of antibiotic storage at -80°C (days)". We have adjusted the data column to reflect this.

We have also added to the footnote of Table 2 to explain that the QC ranges used are the expected MIC ranges for the E. coli strain reported and can be applied to all time points studied.

All amendments are highlighted within the new version of the manuscript.  We thank the reviewers once again for their time and efforts.

This manuscript is a resubmission of an earlier submission. The following is a list of the peer review reports and author responses from that submission.

Round 1

Reviewer 1 Report

Dear Authors, I have following concerns regarding the paper:agent 

A) Use of drug product concentration in aliquot may be more accurate t indicate stability

B) How many samples were tested for each agent (i.e. n=?) 

C) MIC data for IMI, CTX, CTZ, and PTZ show very high variability at Day 179 and do not support your claim of stability upto 6 months.  

Reviewer 2 Report

The authors claim that they had performed a stability test for beta-lactamic antibiotics stored at minus 80ºC. Based on the results of successive antibacterial assays they conclude that the antibiotics are stable for 6 months. The authors did not determine the concentration of the antibiotics over time. In my opinion, this is a major flaw. The stability of beta-lactamics under different conditions had been evaluated countless times and many analytic methods were described. In order to validate the method here pruposed at least a comparison with already estabished methods should had been conducted.